# Analysis of structural and metabolic changes in surface microorganisms following powdery mildew infection in wheat and assessment of their potential function in biological control

Zhen Wu[1], Xiaodong Xue [1,2]*

1 School of Biomedicine and Food Engineering, Shangluo University, Shangluo, Shaanxi, China,
2 Qinling-Bashan Mountains Bioresources Comprehensive Development C.I.C., Hanzhong, Shaanxi, China

* xxd14789632@163.com

## Abstract

Powdery mildew is a highly destructive disease that greatly reduces both the yield and quality of wheat. As there is limited research on changes in microorganism community caused by powdery mildew infection in different tissue parts, especially after spike infection, this study aimed to examine surface microorganisms in infected and healthy wheat plants. Samples were collected from the leaves and spikes, and the number of operational taxonomic units (OTUs), diversity index, abundance, and metabolic changes of the surface microbial community were analysed using 16S rRNA amplicon sequencing technology. Through the identification of surface microbial community in different tissues, 24 phyla were identified in the leaves, and 20 phyla were identified in the spikes. The dominant bacterial phyla observed were *Proteobacteria* and *Bacteroidetes*. At the genus level, 19 genera were detected in the leaves, and 11 genera were detected in the spikes. Notably, the total number of genera in the leaves exceeded that in the spikes. The dominant genera were *Pseudomonas*, *Sphingomonas*, and *Pantoea*. At the species level, there were 37 types identified in leaves and 35 types in spikes. The dominant bacterial species identified included *Pedobacterium panaciterrae*, *Pseudomonas baetica*, *Pseudomonas rhizophaerae*, and *Sphingomonas aerolata*. The analysis conducted in this study revealed that the incidence of powdery mildew was greater in plots situated closer to obstacles than in other plots. Notably, when wheat was infected with powdery mildew, the results indicated that surface microorganisms on both leaves and spikes were significantly impacted, with the response of surface microorganisms on the spikes being more pronounced than that on the leaves. Different from the response of microorganisms on the leaf surface, after infection with powdery mildew, the pathway changes of microorganisms on the spike surface are mainly metabolic regulation. These research results provide theoretical support for the prevention and control of powdery mildew in wheat crops.

**Data availability statement:** The data that support the findings of this study has been uploaded to the SRA under BioProject PRJNA1218831. You can find relevant data in the following link (https://www.ncbi.nlm.nih.gov/search/all/?term=PRJNA1218831).

**Funding:** 1、Shaanxi Provincial Department of Education Key Scientific Research Project (grant no. 21JY008) 2、Shangluo University Science and Technology Research Project (grant no. 20SKY010) 3、General projects of Shaanxi Province's key research and development plan (grant no. 2019NY-067) 4、High-anthocyanin wheat green and efficient cultivation technology integration and demonstration project (grant no. 2023-ZDLNY-13) In this study, supporter Zhen Wu collected the experimental samples, performed the fundamental experimental operations, and authored the initial draft of the paper. Supporter Xiaodong Xue conceptualized the overall experimental design and revised the manuscript.

**Competing interests:** The authors have declared that no competing interests exist.

## 1. Introduction

Common wheat (*Triticum aestivum*) plays a crucial role in ensuring national food security and feeds a significant portion of the global population [1,2]. However, wheat powdery mildew, a global epidemic wheat disease caused by the biotrophic fungus *Blumeria graminis* f. sp. *tritici* (Bgt), can severely reduce wheat yields and affect grain quality [3–5]. Currently, the primary treatment method for controlling the majority of powdery mildews involves the use of chemical agents[6–8]. It is worth mentioning that powdery mildew can induce alterations in the surface microbial community (SMC), leading to the development of resistance among certain microorganisms to powdery mildew [9,10]. Based on this characterization, microbial agents and sterilizing pharmaceuticals resistant to powdery mildew can be identified and developed as an alternative to chemicals agents [11]. Concurrently, powdery mildew can induce metabolic alterations in the microbial community present on the plant's surface, thereby affecting the plant's capacity to respond to environmental stress[12]. It is crucial to analyze the diversity and metabolic changes in the SMC across different parts of the plant following infection with powdery mildew. This analysis is essential for promoting plant growth and enhancing disease resistance.

The most direct application of studying changes in SMC and metabolites is biological control. Biological control of diseased plants is an additional option that does not harm the environment or pollute it. There are many reports on the biological control of powdery mildew [13–16]. Biological control employs antagonistic microorganisms, such as bacteria, as a long-term strategy for managing powdery mildew, in contrast to chemical control methods [17]. Among the biological agents, various antagonistic microorganisms, including those from the *Ampelomyces*, *Bacillus*, *Pseudozyma*, and *Trichoderma*, have demonstrated effectiveness in controlling powdery mildew across a range of crops [18–20]. The metabolites produced by the fungus *Fusarium verticillioides* strain WF18 exhibit significant antifungal activity. These metabolites can effectively inhibit the growth of the pathogen responsible for powdery mildew[21]. Zhou et al. [22] suggested that metabolites produced by endophytic fungi of *Drechslera sp.* have potential biocontrol and bioremediation effects and can effectively inhibit the growth of powdery mildew pathogens. Nevertheless, the availability of microbial species for biological control is limited, with the majority of reports focusing on endophytic bacteria for powdery mildew management. In contrast, there is a scarcity of studies examining surface microorganisms, which poses challenges in meeting both market demands and practical applications. This limited selection of microorganisms is also a significant obstacle to the widespread adoption of biological control, posing a bottleneck in production practices that needs to be addressed.

To increase the effectiveness of biological control against powdery mildew, this study conducted 16S sequencing of the SMC of wheat, with a particular focus on the spike, which has rarely been explored in previous reports. The sequencing analysis revealed significant differences in the SMC between the infected and healthy (control) groups and further identified specific bacteria that have the potential to inhibit powdery mildew. These findings hold great promise for improving the efficacy of biological control methods, reducing reliance on chemical pesticides, and minimizing damage to soil structure and environmental pollution. In addition, this research provides valuable guidance for the scientific and ecological prevention and control of powdery mildew, as well as new insights for plant disease management.

## 2. Materials and methods

### 2.1 Materials

The experimental materials used were the leaves and spikes of wheat (variety: Heijin 3721).

## 2.2 Sample collection and processing

The sampling site is located in the town of Shimen, Luonan County, Shangluo city, Shaanxi Province (E: 110°9' 52.02"; N: 34°12' 23.77"). The sample site size was 100 m × 40 m, with many trees and a low wall (length × height = 400 m × 2.5 m) blocking the ventilation of the area. On the other end, there is a highway. In early May 2023, experimental samples were collected approximately 20 days after wheat heading. The collection method involved dividing the entire plot into three subplots, namely, S1, S2, and S3, and randomly sampling from each subplot. To ensure an adequate sample size for sequencing, we selected 90–100 disease-free plants (referred to as the control, PMX-C) and infected plants (referred to as PMX-S) from each subsample. The plants were further categorized into two types of tissues: leaves and spikes. Samples from each tissue type were randomly mixed and divided into three equal parts (referred to as replicates) and assigned separate numbers (PMX-C1, C2, C3; PMX-S1, S2, S3; PM represents powdery mildew, and X represents leaf or spike) before being sent for sequencing.

## 2.3 Sample surface treatment

Ultrasound was used to extract and collect surface microbial samples from leaves and spikes.

## 2.4 PCR amplification and product recovery

Library construction followed a two-step PCR amplification method. In the first step, specific primers were used to amplify the target fragment, which was then recovered from the gel. In the second step, the recovered product served as a template for secondary PCR amplification. The objective was to incorporate the connectors, sequencing primers, and barcodes necessary for Illumina platform sequencing at both ends of the target fragment. The PCR amplification products were separated and retrieved using 2% agarose gel electrophoresis, the electrophoresis tank was cleaned, and the electrophoresis buffer was replaced. Axygen's AxyPrep DNA gel recovery kit was used for retrieval. Quantitative analysis was conducted using an FTC-3000TM real-time PCR instrument. The samples were combined in an equal molar ratio and subjected to secondary PCR amplification. The necessary connectors for sequencing were added, and the samples were sequenced.

The 16S V3-V4 specific sequences of primers used were as follows:

357F 5'-ACTCCTACGGRAGGCAGCAG-3'
806R 5'-GGACTACHVGGGTWTCTAAT-3'

16S V3-V4 PCR amplification system:

| 5×Buffer | 10μL |
|---|---|
| dNTP (10 mM) | 1μL |
| Phusion Ultra-Fidelity DNA Polymerase | 1μL |
| F/R inner primer (10 uM) | 1μL/1μL |
| Template | 5 ng-50 ng |
| ddH2O | to 50 μL |

16S V3-V4 PCR amplification procedure:

| | 94°C | 2 min |
|---|---|---|
| 38 cycles | 94°C | 30 s |
| | 55°C | 30 s |
| | 72°C | 30 s |
| | 72°C | 5 min |
| | 10°C | Save |

## 2.5 Effective sequence quality control and splicing

The sequencing data were subjected to quality control using Trimmonmatic (version: 0.38) software. The method employed involved removing low-quality reads using a window-based approach. Specifically, for a 50-bp window, if the average quality value within the window was less than 20, the base at the end of the window was truncated. After quality control, reads less than 50 bp in length were filtered out. Next, Cutadapt (version: 1.16) software was used to process the sequencing connectors and primers. Finally, FLASH (version: 1.2.11) software was used for splicing, and paired reads were merged into a single sequence on the basis of their overlap relationship. The minimum overlap length required was 10 bp, and a maximum mismatch ratio of 0.2 was allowed in the overlap area. Nonmatching sequences were subsequently screened.

## 2.6 Sequence optimization and bacterial function prediction

To improve the quality and accuracy of biological information analysis results, it is necessary to perform quality control filtering on the merged reads. Nonspecific amplification caused by PCR amplification can be eliminated by the use of specific primers. The sequence may contain ambiguous bases, homologous regions, and chimaeras generated during the PCR process. Including these sequences in the analysis will reduce the quality of the analysis. Therefore, it is important to remove this portion of the sequence to obtain an optimized sequence for precise analysis. Additionally, it is recommended to filter out singletons (sequences with only one read) in the concatenated long reads, as they are not useful for subsequent clustering of operational taxonomic units (OTUs) data. Furthermore, the OTU annotated species classification results should be removed because they are not relevant to the research objectives. When performing data removal (screening), the following parameters are recommended: maxamplitude=0, maxhomop=8, minlength=200, and maxlength=485. The software Mothur (version: 1.39.5) can be used for this purpose. The optimized sequences were used as data for subsequent OTU and species information analysis.

The functional prediction of bacteria refers to the following website: https://www.kegg.jp/kegg/.

## 3. Results

### 3.1 Analysis of illumina MiSeq sequencing data

In this experiment, a greater incidence of powdery mildew infection was observed on the leaves in subplot S3, which were located in close proximity to the obstacles (Fig 1A). To ensure that the surface bacterial species of the samples remained unchanged, samples were collected from the leaf and spike tissues of wheat in each subplot, and the samples were promptly sent for sequencing. Dilution curves of surface bacteria in various tissues of wheat were generated using high-throughput sequencing on the basis of OTUs. The findings reveal that with increasing number of sequences, the slope of the dilution curve gradually decreases and tends to level off. All the samples eventually reached a plateau stage, indicating that the number of sequences was sufficient to capture the majority of the microbial information in the samples (Fig 1B). In this study, we obtained a total of 67,097 and 67,135 effective sequences for the surface microbiota of wheat leaves and spikes, respectively. After optimization, the number of sequences was reduced to 44,449 and 59,537 for leaves and spikes, respectively. The total number of optimized sequences obtained was 18,860,206 and 25,425,321 for leaves and spikes, respectively. The average length ranges of the optimized sequences for surface bacteria were 237–451 bp and 262–452 bp for leaves and spikes, respectively. The average length of the

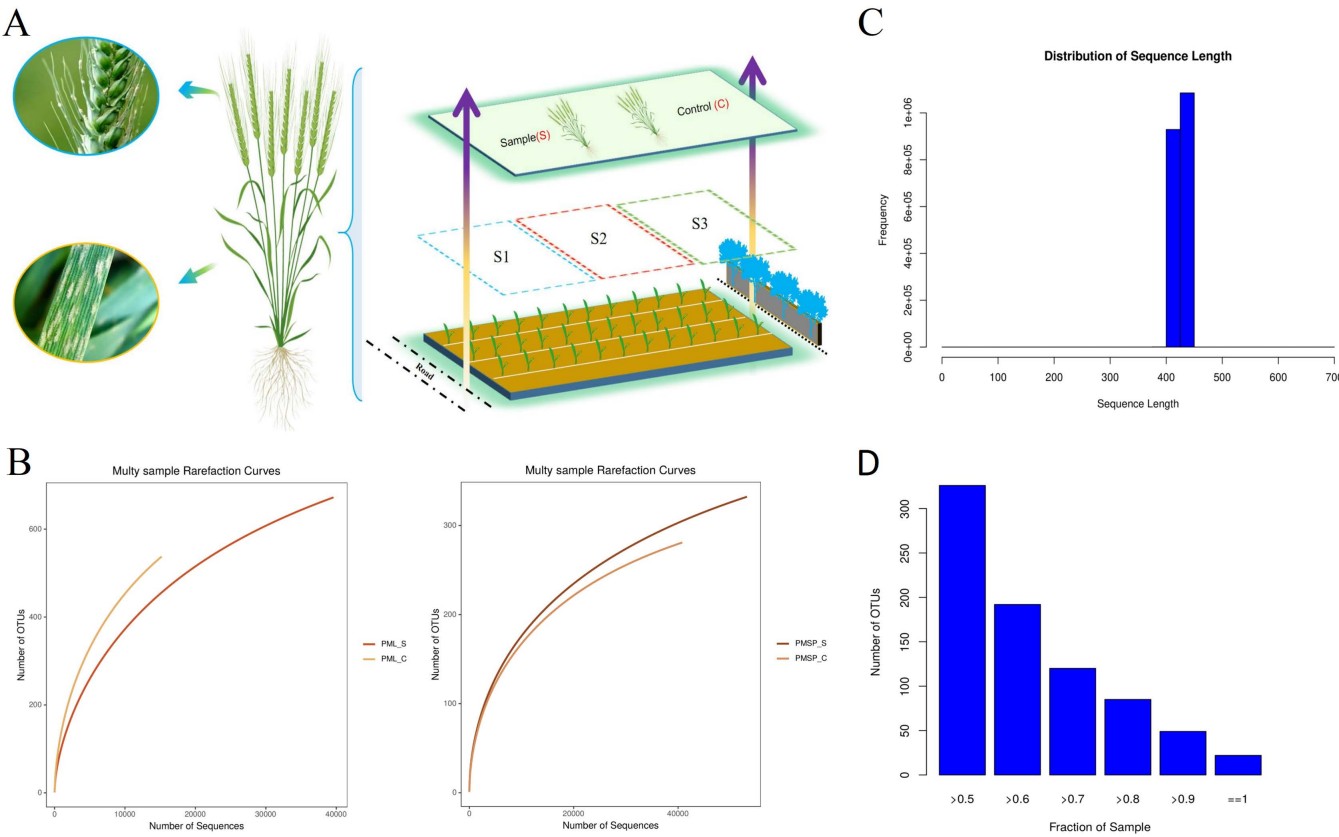

**Fig 1. Sample collection methods and preliminary analysis of sequencing data.** (A) The sample collection method is to divide the entire experimental field into three areas (S1, S2, S3), and collect samples from the control group and experimental group in each area. All samples were analyzed for leaves and spikes. (B) As the number of sequencing increases, the slope of the dilution curve gradually decreases and tends to flatten. PML: indicates leaf, PMSP: indicates spike, _S: infected powdery mildew samples, _C: healthy control sample. (C) The average length of optimized sequences ranges from 400 bp to 450 bp. (D) Relationship between the number of shared OTUs and samples.

optimized sequences was 424 bp and 427 bp for leaves and spikes, respectively (Fig 1C). More than 50% of the samples had 325 identical OTUs (Fig 1D). The range of effective sequence numbers and average length of optimized sequences for all samples is relatively similar, indicating consistent experimental processing.

## 3.2 OTU analysis based on surface microbiota

By analysing the composition of OTUs with a 97% similarity threshold in various samples, we can observe the differences and distances between the samples. Principal coordinate analysis (PCoA) is a method that visualizes the dissimilarities between multiple datasets on a two-dimensional coordinate map. The two axes of the coordinate map represent the two characteristic values that capture the maximum variance. In the PCoA diagram, samples with similar compositions are closer together, indicating a smaller distance between them. PCoA was conducted on the three subplots (S1, S2, and S3) to assess the performance of the leaves and spikes. The analysis revealed that the performance of the leaves differed from that of the spikes. The first principal coordinate, which accounted for 51.9% and 71.1% of the variance in the two tissues, respectively, showed that for the spike, the subplots were significantly different from each other, particularly S2 and S3 (Fig 2A). Powdery mildew had the greatest effect on

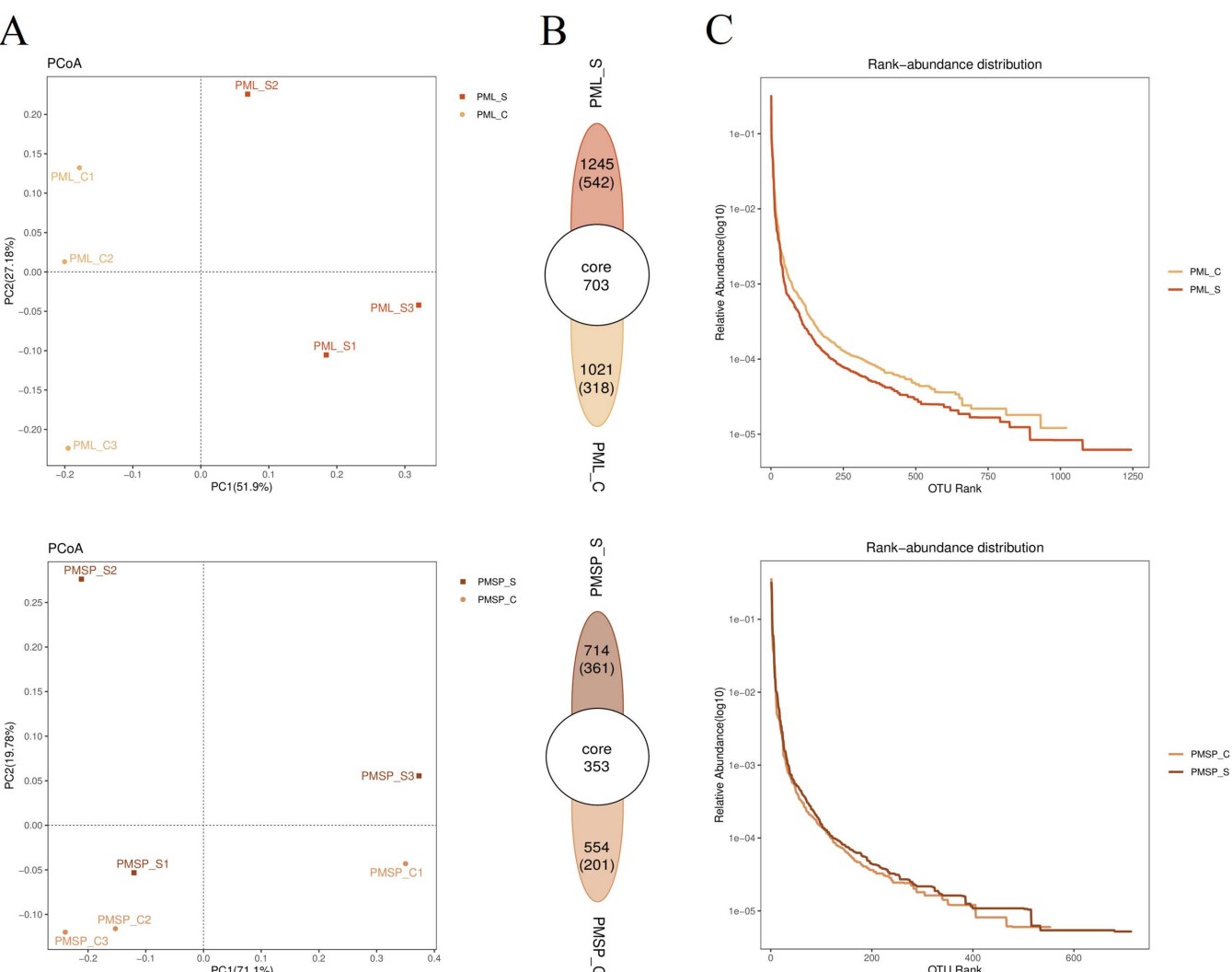

**Fig 2. OTUs analysis based on surface microbiota.** (A) Perform principal coordinate analysis (PCoA) on three subplots (S1, S2, and S3) to evaluate the performance differences leaves and spikes. (B) Analyze the total number of bacterial strains and the number of unique OTUs in the control and experimental group samples. (C) Rank abundance curves were analyzed in order to understand the microbial diversity (species abundance and species evenness) of wheat surface communities. PML: indicates leaf, PMSP: indicates spike, _S: infected powdery mildew samples, _C: healthy control sample.

the spikes, whereas the leaves were minimally affected. Owing to the small overall differences among the three subsample plots and the limited publication space, this study focused on conducting a comparative analysis between infected and noninfected samples (controls) in subsequent analyses, without distinguishing between sample plots.

At a similarity level of 97%, the number of OTUs in the surface microbiota of susceptible wheat samples was 1245 in the leaf and 714 in the spike (Fig 2B); similarly, 703 and 353 OTUs were shared with the noninfected (control) samples, respectively. Additionally, there were 542 and 361 unique OTUs in the infected leaf and spike samples, accounting for 43.53% and 50.56% of the total OTUs, respectively. The total number of bacterial strains and the number of unique OTUs in the infected samples were greater than those in the noninfected samples. These findings demonstrate that the diversity of the surface microbiota in various tissues was greater than that in the control group. This study provides evidence that the diversity

of bacterial communities in infected tissues is significantly greater than that in noninfected tissues. Additionally, the number of unique OTUs in the spike was greater, with each OTU corresponding to a bacterial species. This indicates a greater alteration in bacterial species on the surface of the spike following infection.

The rank abundance curve is a useful tool for understanding microbial diversity in terms of species abundance and species evenness. The width of the curve represents the abundance of a species, with a larger range on the horizontal axis indicating greater species abundance. The smoothness of the curve reflects the evenness of the species distribution in the sample, with a flatter curve indicating a more uniform distribution. In this study, the abundance levels of various tissues (Fig 2C) were analysed. Compared with those of noninfected samples, the curves of infected samples presented a greater range on the horizontal axis, indicating greater species abundance. Additionally, the curves of the spike samples were slightly smoother than those of the noninfected samples, suggesting better species evenness. Therefore, compared with the noninfected samples, the infected samples presented both greater species abundance and greater species evenness in the spikes.

### 3.3 Analysis of surface bacterial species composition

To investigate the composition and structure of the surface microbiota in various tissues of wheat (leaves and spikes), a comparative analysis was conducted at three levels: phylum, genus, and species. At the phylum level, 24 phyla were associated with the leaves, and 20 phyla were associated with the spikes (Fig 3A). The top four dominant bacteria in these phyla are *Proteobacteria*, *Bacteroidetes*, *Actinobacteria* and *Firmicutes*, and *Proteobacteria* had the highest abundance in leaves and spikes, accounting for 83.04% and 88.47%, respectively. *Bacteroidetes* had the second highest abundance, accounting for 9.00% and 3.35%, respectively. *Actinobacteria* ranked third in terms of leaf abundance, accounting for 4.63%, whereas *Firmicutes* ranked third in terms of spike abundance, accounting for 5.72% (Table 1). However, there were no significant differences observed between the susceptible group and the control group for several other phyla in different tissues. Only *Firmicutes* exhibited a significant difference in the spikes, which was approximately 19 times greater than that of the control (0.05716/0.00303).

At the genus level, the leaves presented 19 genera, and the spikes presented 11 genera when the detected abundance was greater than 0.5% (Fig 3B). The total number of leaf genera was greater than that of spike genera. The dominant genera in the leaves were *Pantoea* (33.46%), *Pseudomonas* (14.80%), *Massilia* (10.77%), and *Sphingomonas* (10.39%). In the spike, the dominant genera were *Pantoea* (32.76%), *Pseudomonas* (24.65%), *Massilia* (20.99%), and *Anoxybacillus* (4.38%). *Pseudomonas* was highly abundant in all the tissues. Furthermore, when the proportions of leaf and spike components at the genus level were considered, the first three items with the highest abundance exhibited a certain resemblance, possibly because they both belong to the aboveground parts.

At the species level, the top 50 species were analysed, including unclassified and uncultured bacterial species (Fig 3C). The dominant species in the leaves were as follows: *Pseudomonas rhizophaerae* (8.81%), *Sphingomonas aerolata* (7.31%), *Rahnella aquatilis* (6.17%), and *Pseudomonas baetica* (4.00%). The dominant species in the spike were *P. rhizophaerae* (22.93%), *S. aerolata* (2.33%), *Duganella zoogloides* (1.52%), and *P. baetica* (1.02%).

### 3.4 Differences in microbial communities between infected and noninfected plants

A comparison of infected and noninfected samples from different tissues revealed that the overall difference was not significant (Fig 4A). A differential analysis was subsequently

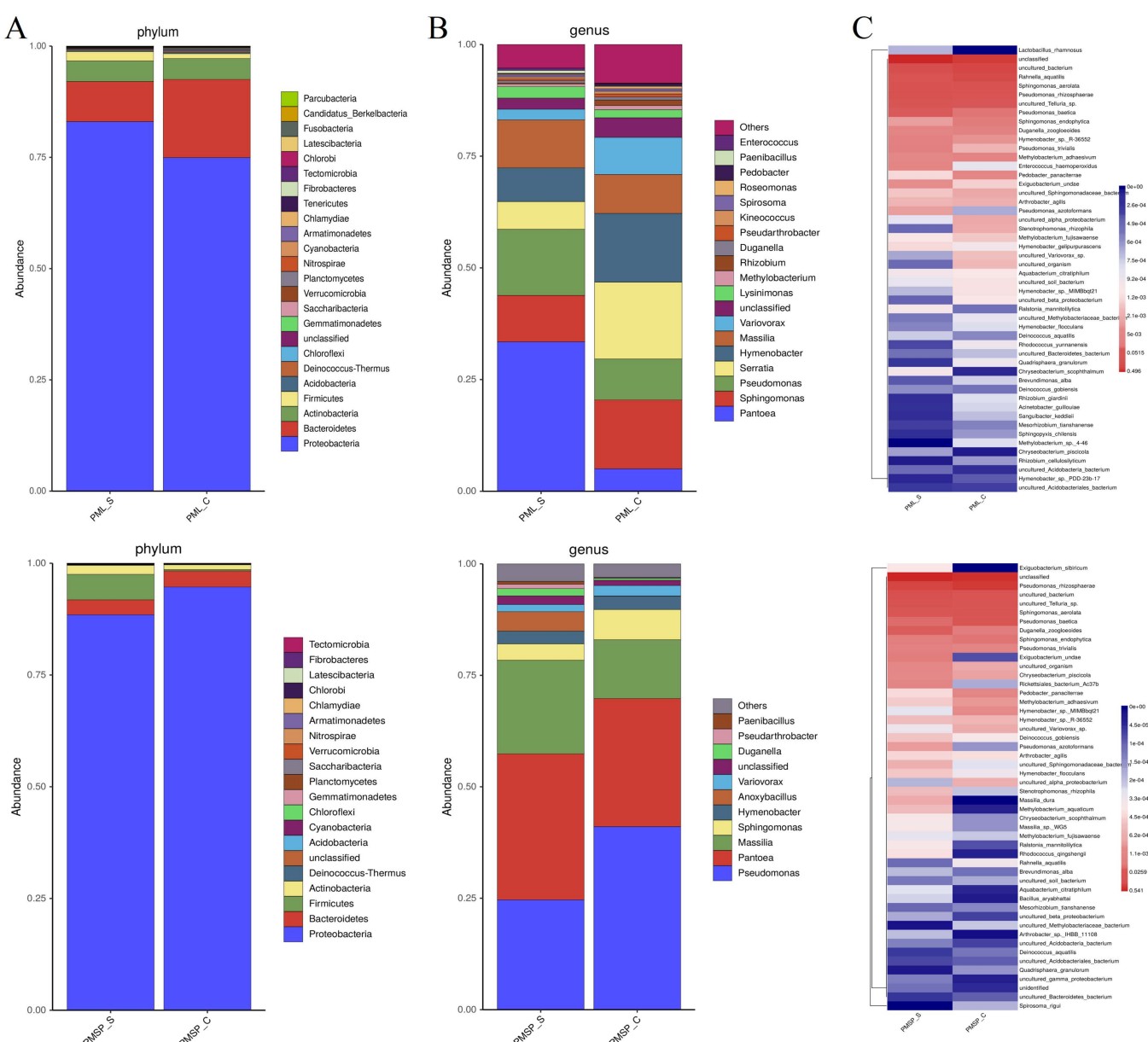

**Fig 3. Analysis of bacterial species composition on wheat surface.** To investigate the composition and structure of surface microbiota in various tissues of wheat, a comparative analysis was conducted at three levels: phylum (A), genus (B), and species (C). PML: indicates leaf, PMSP: indicates spike, _S: infected powdery mildew samples, _C: healthy control sample.

conducted on the species levels of different tissue samples. The highly significant differences between the leaf samples were related to *Pseudomonas trivialis* (an aerobe mesophilic bacterium that was isolated from the phyllosphere of grasses) and *Pseudomonas azotoformans* (a bacterium that has the ability to suppress plant diseases by protecting the seeds and roots from fungal infection). This ability is due to secondary metabolites produced by these bacteria, such as antibiotics, siderophores, and hydrogen cyanide, as well as the ability of these bacteria to rapidly colonize the rhizosphere and outcompete some pathogens [23–26]. The differences that were generally significant are *Sphingomonas endophytica* (an aerobe, mesophilic,

**Table 1. Distribution of surface microorganisms at phylum level in leaf and spike samples.**

|  | Proteobacteria | Bacteroidetes | Actinobacteria | Firmicutes | Acidobacteria |
|---|---|---|---|---|---|
| PML_S | 0.83038 | 0.08999 | 0.04634 | 0.02089 | 0.00339 |
| PML_C | 0.74943 | 0.17563 | 0.04718 | 0.01075 | 0.00350 |
|  | Proteobacteria | Bacteroidetes | Firmicutes | Actinobacteria | Deinococcus-Thermus |
| PMSP_S | 0.88471 | 0.03349 | 0.05716 | 0.01986 | 0.00107 |
| PMSP_C | 0.94691 | 0.03593 | 0.00303 | 0.01055 | 0.00092 |

PML: indicates leaf. PMSP: indicates spike. _S: infected powdery mildew samples. _C: healthy control sample.

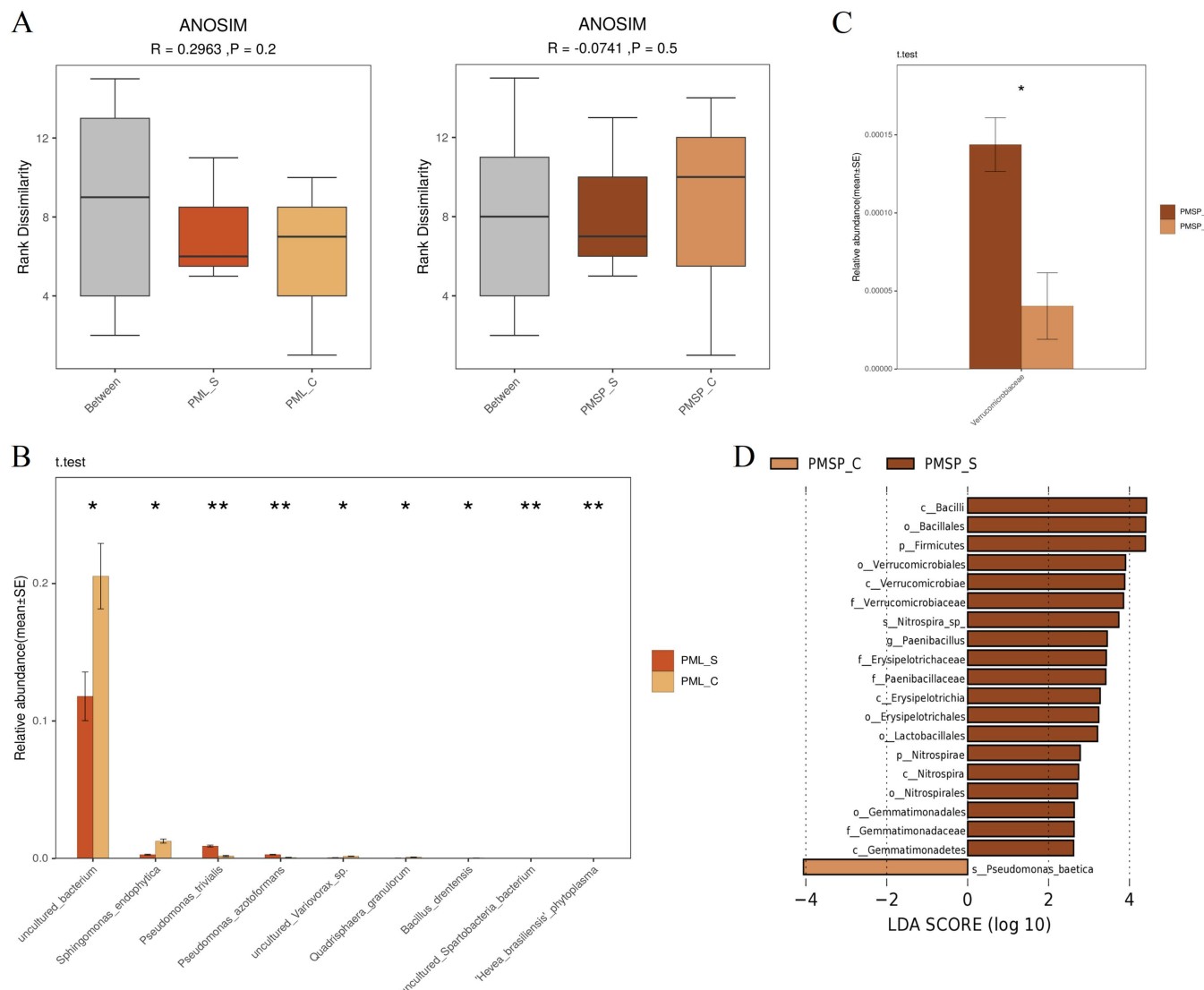

**Fig 4. Differences in surface microbial communities between infected and uninfected plants.** (A) Anosim analysis was performed to analyze the differences between the species levels of the control and experimental groups. Anova_Ttest parametric test was performed to analyze the differences in the surface microbial community of samples from wheat leaves (B) and spikes (C). (D) LDA analysis was performed on the spike samples to find the bacterial species that caused the differences between the control and experimental group samples. PML: indicates leaf, PMSP: indicates spike, _S: infected powdery mildew samples, _C: healthy control sample.

gram-negative bacterium that was isolated from surface-sterilized roots of Artemisia annua), uncultured *Variovorax sp.* (which has been cultivated and whose respiration includes deha-lorespiration and iron reduction), and *Quadrisphaera granulorum* (an aerobe, mesophilic, gram-positive bacterium that was isolated from aerobic granule sludge biomass) (Fig 4B).

At the species level, there was no significant difference between the bacterial species in the infected and control spike samples; only slight differences were observed at the family level and above; the main difference in the family was *Verrucomicrobiaceae* (*P*=0.0211) (Fig 4C), which is found in a wide variety of environments [27] and is thought to have soil remediation capabilities.

The linear discriminant analysis (LDA) effect size software LEfSe is used in the discovery of high-dimensional biomarkers and the identification of genomic features. It is specifically designed to distinguish between two or more biological conditions or groups. LEfSe uses linear discriminant analysis (LDA) to estimate the impact of each component's (species') abundance on the differential effect. The LDA results revealed significant differences among bacterial species in various tissues. Specifically, 23 species were detected in the leaves, and 19 species were detected in the spikes (Fig 4D). In the leaves, the bacterial species with the greatest contribution was *g_Pantoea* (The single letters in front of "_" respectively represent the abbreviations of kingdom, phylum, class, order, family, genus and species, the same below.). In the spikes, the bacterial species with the greatest contribution were *c_Bacilli*, *o_Bacillales*, and *p_Firmicutes*, and the other categories with significant impacts that were relatively smooth decreased in order. The figure shows that the microbial community in the control sample of the spike was influenced primarily by a single item (*s_Pseudomonas baetica*), whereas that in the leaf samples were influenced by multiple categories.

## 4.  Discussion

### 4.1  Analysis of the influence of the local environment on powdery mildew

The environment has a significant effect on microorganisms, including pH, temperature, humidity, and light factors, among others [28–31]. In the case of powdery mildew, the quality of ventilation also directly affects the severity of the disease [32]. Therefore, studying the influence of obstacles on the occurrence of powdery mildew during wheat growth is highly important. However, there are currently limited research reports available on this topic. This article categorized the experimental plots based on the characteristics of their surrounding environment. Plot S1 was located approximately 10 metres away from the highway, plot S3 was closer to the low wall and trees at a distance of approximately 7 metres, and plot S2 was situated between S1 and S3 (Fig 1A). The analysis conducted in this study revealed that the incidence of powdery mildew was greater in plots closer to obstacles than in other plots. Specifically, the number of powdery mildew spots in the S3 plots was greater than that in the S1 and S2 plots, particularly in the spikes. However, the difference between S1 and S2 is not significant (Fig 2A). During the growth process of wheat, nearby obstacles can influence its ventilation, resulting in the occurrence of powdery mildew. The spikes are the most affected, followed by the leaves. However, owing to the abundance of leaves, with many overlapping and varying distances between individuals, the impact of ventilation on powdery mildew is at an lower level. Only the spike possesses ample space between individuals, and the degree of ventilation directly affects the speed of air circulation. Therefore, spikes are the most significantly affected by ventilation conditions.

### 4.2  Analysis of bacterial communities on the surfaces of different wheat tissues

On the basis of the results of the OTU analysis (3.2) and surface bacterial composition analysis (3.3), it can be inferred that the SMC on the spikes was smaller than that on leaves. The LDA

analysis revealed that there were no significant differences between the infected and control samples at the species level, except for minor variations at the family and higher taxonomic levels. This suggests that only a few bacterial communities on the spike undergo changes following infection, with minimal alterations at the fundamental level.

## 4.3 Analysis of surface microbial changes and potential for disease prevention

The functional analysis of differential bacterial communities between the susceptible group and the control group is crucial for comprehending the dynamic changes in disease occurrence and predicting disease severity. This information provides an important basis for understanding the biological mechanism of disease occurrence and carrying out biological control [33–36]. The distinction between susceptible samples and control samples in different tissue parts forms the theoretical foundation of biological control [37]. Given the high proliferation of bacteria in susceptible samples, targeted prevention and control measures can be implemented to combat these microorganisms. In this case, the synergistic effect between microorganisms may lead to the eradication of pathogenic microorganisms in susceptible samples. Conversely, if there is high proliferation of bacteria in the control sample, these microorganisms can be further amplified, potentially representing a crucial bacterial group for inhibiting diseases such as powdery mildew. However, many of these microorganisms are challenging to culture under laboratory conditions. Traditional microbial research methods that rely on pure culture cannot adequately reveal the differences and functions of microbial community structures present on the surface of wheat tissues. High-throughput sequencing addresses the limitations of these traditional methods and significantly enhances the amount of information available regarding unknown microorganisms. It serves as an effective analytical approach.

To connect the bacterial community with its potential functions, this study employed PICRUSt2 for a comparative analysis based on the KEGG database, utilizing 16S sequencing data obtained from both the control and susceptible groups. This approach aimed to predict the potential functions of the bacterial community present on the surface of wheat plants. The results revealed 19 metabolic pathways that exhibited significant differences in the leaves. The three most abundant pathways identified were ko01100, ko02020, and ko00650. Among these, the upregulated pathways included ko04659, ko04914, ko05215, ko04915, ko04657, ko04612, ko02025, ko02020, ko00250, and ko04217, whereas the remaining pathways were downregulated (Fig 5A). The functions of the 19 pathways are primarily organized into six categories: biosynthesis of other secondary metabolites, amino acid metabolism, energy metabolism, cellular processes, organismal systems, and human diseases. In comparison, the third-level pathways predominantly included the methylaspartate cycle, undecylprodigiosin biosynthesis, glutathione biosynthesis, ketone body biosynthesis, (gamma-Aminobutyrate) shunt, pyrimidine degradation, archaea, central carbohydrate metabolism, leucine biosynthesis, cell growth and death, the IL-17 signalling pathway, Th17 cell differentiation, and Chemical carcinogenesis (S1 and S2 Tables). Among these pathways, the methylaspartate cycle serves as an alternative route for converting acetyl-coenzyme A (acetyl-CoA) into biosynthetic intermediates without the concomitant loss of carbon dioxide. In surface microorganisms of plants, this cycle can facilitate the utilization of organic substrates such as fatty acids, alcohols, and esters, particularly, when these microorganisms are unable to employ glyoxylate bypass. These findings indicate that the occurrence of wheat powdery mildew may affect the growth and metabolism of some microorganisms. The biosynthesis of undecylprodigiosin and glutathione are pathways activated by microbial interactions, with a particular emphasis on glutathione biosynthesis. An increase in glutathione can mitigate oxidative stress caused by powdery mildew

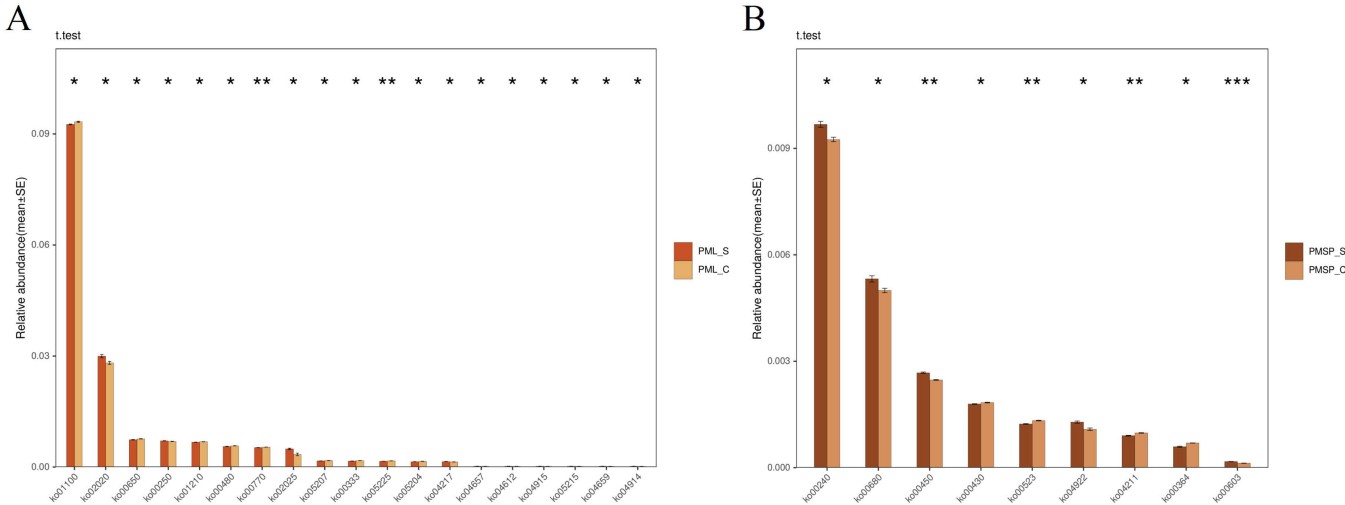

**Fig 5. Pathway analysis with significant differences between infected samples and control samples.** A comparative analysis using the KEGG pathways database was conducted on leaf (A) and spike (B) samples to identify pathways with significant differences between infected samples and control samples. PML: indicates leaf, PMSP: indicates spike, _S: infected powdery mildew samples, _C: healthy control sample.

[38]. Additionally, changes in pyrimidine degradation and central carbohydrate metabolism are associated with microbial metabolic adjustments and alterations in energy supply. Ketone body biosynthesis is linked to energy metabolism and the utilization of fatty acids, particularly under adverse conditions. It has been reported that following wheat infection by powdery mildew, the corresponding defence response signalling pathway is activated to counteract the threat posed by the disease [39].

The gamma-aminobutyrate shunt, leucine biosynthesis, and cell growth and death pathways are crucial for activating protein synthesis and regulating metabolism in wheat following infection. These pathways play significant roles in managing microbial stress responses, coordinating defence mechanisms against various microorganisms, and enhancing overall disease resistance [40,41]. However, the IL-17 signalling pathway, Th17 cell differentiation, and chemical carcinogenesis primarily influence immune responses in animals. Analogous signalling mechanisms that also impact the immune responses of microorganisms may exist. Additionally, alterations in pathways associated with archaea may indicate modifications in the structure of the microbial community on the plant surface, which could influence the metabolic activities of specific archaea, resulting in changes in gene expression or enzyme activity within related pathways. Currently, there are no published literature reports addressing these findings.

Nine metabolic pathways exhibited significant differences in the spikes, with the three most abundant pathways being ko00240, ko00680, and ko00450. Among the nine, the upregulated pathways included ko00240, ko00680, ko00450, ko04922, and ko00603. Notably, the three pathways with the highest abundance were all upregulated, and the overall trend of change among the nine pathways was more gradual than that of the microorganisms on the leaf surface (Fig 5B). The primary pathways and functions identified are energy metabolism, organismal systems, glycan metabolism, carbohydrate metabolism, and the metabolism of other amino acids. This finding indicates that, following infection with powdery mildew, the pathway changes in surface microorganisms associated with the spike protein primarily reflect metabolic adjustments. Among the three-level pathways associated with energy metabolism, those that were upregulated include methanogenesis, coenzyme M biosynthesis, methane

oxidation, formaldehyde assimilation, the acetyl-CoA pathway, and serine biosynthesis. The upregulation of these pathways suggests that following infection with powdery mildew, certain aspects of the surface microbial community (SMC) experience increased metabolic activities, which may be linked to plant defence responses [42]. Concurrently, the increase in energy metabolism may facilitate the synthesis of disease-resistant substances, provide energy for cellular growth and repair, and help maintain normal cellular functions. In contrast, the phosphate acetyltransferase-acetate kinase pathway was downregulated. This pathway plays a crucial role in microbial carbon and energy metabolism, indicating a shift in the metabolism of surface microorganisms. The relative abundance of some microbial species that previously relied on this pathway may have decreased, whereas that of other species employing different metabolic strategies may have increased. This alteration in community structure could impact the functionality of the entire microbial community, thereby influencing wheat resistance to powdery mildew [43,44]. Additionally, among the third-level pathways under organismal systems, the glucagon signalling pathway is upregulated. Microorganisms utilize this pathway to produce signalling molecules or metabolites that influence the physiological state and defence responses of wheat, such as inducing systemic resistance and enhancing overall resistance to powdery mildew [45]. The longevity-regulating pathway is downregulated, and currently, there are no reports elucidating the role of this pathway. It is possible that certain microorganisms diminish their investment in mechanisms associated with long-term survival and maintenance of their cellular lifespan, opting instead to prioritize short-term growth and reproduction. This shift may arise from changes in the surface environment of wheat caused by powdery mildew infection, prompting microorganisms to rapidly increase their populations in a short timeframe to better adapt to or exploit new environmental conditions. The observed increase in glycan metabolism, carbohydrate metabolism, and the metabolism of other amino acids suggests that following wheat infection with powdery mildew, the metabolic activity of surface microorganisms on the spike is increased, leading to increased energy demands that bolster resistance against pathogenic bacteria. This overall increase in plant defence mechanisms aligns with findings in bacteria[46–48]. Notably, when wheat is infected with powdery mildew, the surface microorganisms on the spike respond more drastic than those on the leaves do; although the relative abundance is lower, the difference is significant, and the number of responding species is also relatively small. This phenomenon may be attributed to the relatively simple structure of the microbial community present in the spike. Overall, there is a tendency to enhance metabolic pathways related to sugar and energy metabolism, thereby establishing a protective barrier for the entire spike, improving wheat disease resistance, and mitigating the threat posed by disease. This may reflect the plant's evolutionary strategy to safeguard its offspring (seeds), indicating a protective mechanism that has coevolved with microorganisms.

Through metabolite analysis, we can gain a deeper understanding of the functions of metabolites in disease prevention and control. An analysis of the microorganisms on the leaf surface via three-level pathway analysis revealed that many microorganisms have potential disease control functions. For example, the biosynthesis of prodigiosin, which is typically associated with Shiga toxin-producing *Escherichia coli* (STEC), *Shigella*, and *Bacillus amyloliquefaciens*, produces a secondary metabolite with notable antibacterial activity. This compound shows promise for applications in microbial disease control by inhibiting the growth of other microorganisms. Similarly, undecylprodigiosin, which is produced primarily by *Streptomyces spp.*, *Serratia marcescens*, and *Pseudoalteromonas*, is another secondary metabolite with antibacterial properties that can contribute to the prevention and control of microbial diseases. Furthermore, glutathione, which is commonly produced by *Streptococcus agalactiae*, *Saccharomyces cerevisiae*, and *Pasteurella multocida*, is an

important antioxidant found in both plants and microorganisms. It plays a crucial role in combating oxidative stress and disease. Additionally, the GABA (gamma-aminobutyrate) shunt, typically produced by *Bacillus thuringiensis*, *Lactobacillus brevis*, is related to plant stress resistance. This pathway can regulate the plant's response to stress, including disease prevention and treatment. Pantothenate, typically produced by *Salmonella typhimurium*, *Escherichia coli*, *Corynebacterium glutamicum*, and *Clostridium*, is a crucial component of coenzyme A. biosynthesis is essential for the growth and metabolism of microorganisms and plays a significant role in microbial disease control by influencing the metabolic pathways of these organisms. Biofilm formation, associated with *Pseudomonas aeruginosa*, is linked to various microbial diseases. A better understanding and control of biofilm formation could aid in the prevention and treatment of these diseases. Antigen processing and presentation, which are commonly associated with *Pseudomonas*, *Bacillus subtilis*, and *Escherichia coli*, are integral to the host immune response to microbial infections and contribute to the prevention and control of microbial diseases by enhancing the host immune response. Intriguingly, the IL-17 signalling pathway and Th17 cell differentiation are involved primarily in animal immune responses. However, similar signalling mechanisms may also play a role in the defence of microorganisms against pathogens, although there are currently no reports documenting this role.

During the analysis of spike surface microorganisms, numerous microorganisms with potential disease prevention and control functions have been identified. For example, fluorobenzoate degradation, typically by *P. aeruginosa*, *Pseudomonas putida*, and *Alcaligenes faecalis*, is associated with organic pollution. The degradation of such chemicals is linked to the tolerance of plants to environmental pollutants, thereby indirectly influencing the prevention and control of plant diseases. Additionally, the metabolism of selenocompounds, which are commonly attributed to *A. faecalis*, *Bacillus licheniformis*, and *Thiobacillus ferrooxidans*, plays a role in plant antioxidant defence. Selenium serves as an important antioxidant that protects plants from oxidative stress, which may be related to disease control. Furthermore, glycosphingolipid biosynthesis, specifically the globo-series, usually occurs in *Glomeromycota fungi*, *Pseudomonas aeruginosa*, and *Bifidobacterium longum*. The biosynthesis of glycosphingolipids is linked to the recognition of pathogens, thereby affecting disease prevention and control. The acetyl-CoA pathway, which is primarily observed in autogenous nitrogen-fixing bacteria, methanotrophs, and Acetobacter acetiformis, leads to the synthesis of numerous secondary metabolites that exhibit antibacterial, anti-insect, and other defence functions. This pathway supplies essential precursors for plants to produce defence-related substances, thereby playing a significant role in the prevention and control of plant diseases. It helps plants generate disease-resistant compounds to counteract pathogenic bacteria and other threats when facing disease. Similarly, serine biosynthesis, typically facilitated by *Enterobacter aerogenes*, *E. coli*, and *Pseudomonas putida*, is crucial for plants, as it produces important amino acids involved in protein synthesis and various metabolic pathways. In the context of plant disease control, serine serves as a precursor for the synthesis of other defence-related metabolites. Additionally, the production of certain disease-resistant proteins in plants necessitates the involvement of serine. Consequently, this pathway indirectly supports plants in their response to diseases by ensuring a sufficient supply of serine, thereby contributing to defence responses associated with plant disease control.

The results of this study indicate that following the infection of wheat by powdery mildew, significant alterations in the structure, metabolism, and cellular regulation processes of microorganisms present on the surfaces of leaves and spikes occur. During this process, many microorganisms exhibit disease resistance, providing a crucial foundation for the prevention and control of plant diseases, as well as for the development of related microbial pesticides

and fungicides. Additionally, during the sampling conducted in this study, few leaves were infected with powdery mildew, and the density of powdery mildew pathogens on individual leaves was low. Overall, this limited infection did not result in severe damage to the wheat plants. Consequently, the abundance of SMC remained relatively high, and their diversity increased. However, in cases of severe infection, where entire leaves are extensively covered with powdery mildew conidia and mycelia, these pathogens are dominant on the leaf surface, which significantly affects the structure and metabolism of surface microorganisms. In this case, if beneficial microorganisms for disease control are screened, the effect will be better. This is a limitation of this study, and we will continue to research this issue in depth in the future.

## Supporting information

**S1 Table. KEGG pathway analysis of microorganisms on the surface of leaves.** (XLSX)

**S2 Table. KEGG pathway analysis of microorganisms on the surface of spikes.** (XLSX)

## Acknowledgements

We are grateful to Professor Bingde Dou (Shaanxi Institute of Biological Agriculture) and Shengbao Xu (North West Agriculture and Forestry University) for providing the wheat seed material and technical guidance.

## Author contributions

**Conceptualization:** Xiaodong Xue.

**Formal analysis:** Zhen Wu.

**Funding acquisition:** Xiaodong Xue.

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
