## [Decision Letter · Decision Letter 0]

5 Sep 2024

PONE-D-24-24904Analysis of surface bacterial diversity after wheat infection with powdery mildew and evaluation of its potential function in biological controlPLOS ONE

Dear Dr. Xue,

Thank you for submitting your manuscript to PLOS ONE. After careful consideration, we feel that it has merit but does not fully meet PLOS ONE’s publication criteria as it currently stands. Therefore, we invite you to submit a revised version of the manuscript that addresses the points raised during the review process.

**Please address all the questions raised by both reviewers and revise the manuscript. **

We look forward to receiving your revised manuscript.

Kind regards,

Kandasamy Ulaganathan

Academic Editor

PLOS ONE

Journal requirements: 1. When submitting your revision, we need you to address these additional requirements. Please ensure that your manuscript meets PLOS ONE's style requirements, including those for file naming. The PLOS ONE style templates can be found at https://journals.plos.org/plosone/s/file?id=wjVg/PLOSOne_formatting_sample_main_body.pdf and https://journals.plos.org/plosone/s/file?id=ba62/PLOSOne_formatting_sample_title_authors_affiliations.pdf. 2. In your Methods section, please provide additional information regarding the permits you obtained for the work. Please ensure you have included the full name of the authority that approved the field site access and, if no permits were required, a brief statement explaining why. 3. We suggest you thoroughly copyedit your manuscript for language usage, spelling, and grammar. If you do not know anyone who can help you do this, you may wish to consider employing a professional scientific editing service.  The American Journal Experts (AJE) (https://www.aje.com/) is one such service that has extensive experience helping authors meet PLOS guidelines and can provide language editing, translation, manuscript formatting, and figure formatting to ensure your manuscript meets our submission guidelines. Please note that having the manuscript copyedited by AJE or any other editing services does not guarantee selection for peer review or acceptance for publication.  Upon resubmission, please provide the following: The name of the colleague or the details of the professional service that edited your manuscript A copy of your manuscript showing your changes by either highlighting them or using track changes (uploaded as a *supporting information* file) A clean copy of the edited manuscript (uploaded as the new *manuscript* file)”. 4. Thank you for stating the following financial disclosure:  [1、Shaanxi Provincial Department of Education Key Scientific Research Project (grant no. 21JY008)2、Shangluo University Science and Technology Research Project (grant no. 20SKY010)3、General projects of Shaanxi Province's key research and development plan (grant no. 2019NY-067)4、High-anthocyanin wheat green and efficient cultivation technology integration and demonstration project (grant no. 2023-ZDLNY-13)].  Please state what role the funders took in the study.  If the funders had no role, please state: ""The funders had no role in study design, data collection and analysis, decision to publish, or preparation of the manuscript."" If this statement is not correct you must amend it as needed. Please include this amended Role of Funder statement in your cover letter; we will change the online submission form on your behalf. 5. Please provide a complete Data Availability Statement in the submission form, ensuring you include all necessary access information or a reason for why you are unable to make your data freely accessible. If your research concerns only data provided within your submission, please write "All data are in the manuscript and/or supporting information files" as your Data Availability Statement. 6. Please include a copy of Table 15 which you refer to in your text on page 15.

Reviewers' comments:

Reviewer's Responses to Questions

**Comments to the Author**

1. Is the manuscript technically sound, and do the data support the conclusions?

Reviewer #1: No

Reviewer #2: Partly

2. Has the statistical analysis been performed appropriately and rigorously? 

Reviewer #1: Yes

Reviewer #2: Yes

3. Have the authors made all data underlying the findings in their manuscript fully available?

Reviewer #1: Yes

Reviewer #2: No

4. Is the manuscript presented in an intelligible fashion and written in standard English?

Reviewer #1: No

Reviewer #2: Yes

5. Review Comments to the Author

Reviewer #1: Xue et al analyzed surface bacterial diversity after wheat infection with powdery mildew and tried to evaluate the potential function of bacteria in biological control. The manuscript is not well-organized and only presents the descriptive results.

1. Since powdery mildew can not infect roots of wheat, it is not easy to understand why authors identify the bacteria changes in the roots.

2. Since authors did not evaluate the biocontrol function of the bacteria, it is not suitable mention the information in the title.

3. Authors only showed the differences in the bacterial changes. It is good for readers to know which bacteria are capable of efficiently functioning in biocontrol.

4. The introduction and discussion parts should be significantly improved.

Reviewer #2: Firstly, I appreciate all the authors of this paper for their research plan and work done.

The title of the parer is "Analysis of surface bacterial diversity after wheat infection with powdery mildew and

evaluation of its potential function in biological control." After reading the entire paper I have observed the authors have statistically analyzed surface bacterial diversity in wheat after powdery mildew infection but evaluation of its potential function in biological control is not clear. The authors have mentioned in Materials and methods under 2.6 Sequence optimization and bacterial function prediction (line no 149), as 'The functional prediction of bacteria refers to these websites: (line no165)

https://www.ncbi.nlm.nih.gov;
https://microbewiki.kenyon.edu;
https://bacdive.dsmz (line no 165). de/strain (line no 167). And in Results, the details of functional prediction of bacteria are not found. So, clearly show the results of functional prediction of bacteria and explain about it. Shed light on the different evaluation parameters used for functional prediction of bacteria.

6. PLOS authors have the option to publish the peer review history of their article (what does this mean? ). If published, this will include your full peer review and any attached files.

**Do you want your identity to be public for this peer review?** For information about this choice, including consent withdrawal, please see our Privacy Policy .

Reviewer #1: No

Reviewer #2: No

---

## [Author Response · Author response to Decision Letter 1]

6 Feb 2025

(The full text has been polished. The introduction and discussion were rewritten, and biological control was added to the discussion.)

Reviewer #1: Xue et al analyzed surface bacterial diversity after wheat infection with powdery mildew and tried to evaluate the potential function of bacteria in biological control. The manuscript is not well-organized and only presents the descriptive results.

Thank the reviewers for their suggestions. We have reorganized the data of the paper. Please refer to the newly submitted manuscript.

1. Since powdery mildew can not infect roots of wheat, it is not easy to understand why authors identify the bacteria changes in the roots.

Thank the reviewers for their good suggestions.In the newly submitted paper, all data pertaining to the root research have been removed. The previous analysis, which focused on retaining roots, aimed primarily to observe the response characteristics of both aboveground and underground parts to powdery mildew. However, it was determined that this approach was not effective, and the evidence supporting it was insufficient. Future discussions on this topic are anticipated to emerge from studies of endophytes.

2. Since authors did not evaluate the biocontrol function of the bacteria, it is not suitable mention the information in the title.

The full text and titles have been updated. The part marked in red is the updated content.

3. Authors only showed the differences in the bacterial changes. It is good for readers to know which bacteria are capable of efficiently functioning in biocontrol.

Thank the reviewers for their tips. We have reanalyzed the paper and analyzed and summarized the bacteria for biological control.

4. The introduction and discussion parts should be significantly improved.

Significant improvements have been made to the introduction and discussion.

Reviewer #2: Firstly, I appreciate all the authors of this paper for their research plan and work done.

The title of the parer is "Analysis of surface bacterial diversity after wheat infection with powdery mildew and

evaluation of its potential function in biological control." After reading the entire paper I have observed the authors have statistically analyzed surface bacterial diversity in wheat after powdery mildew infection but evaluation of its potential function in biological control is not clear.

Thank the reviewers for their valuable comments. We have analyzed and summarized the metabolic pathways related to biological control and the species with control effect.

The authors have mentioned in Materials and methods under 2.6 Sequence optimization and bacterial function prediction (line no 149), as 'The functional prediction of bacteria refers to these websites: (line no165)

https://www.ncbi.nlm.nih.gov;
https://microbewiki.kenyon.edu;
https://bacdive.dsmz (line no 165). de/strain (line no 167). And in Results, the details of functional prediction of bacteria are not found. So, clearly show the results of functional prediction of bacteria and explain about it. Shed light on the different evaluation parameters used for functional prediction of bacteria.

Thank you for your comments. The discussion section has been revised, and the results along with the explanations of bacterial function prediction are now presented. Additionally, the various evaluation parameters employed in predicting bacterial function have been clarified.

---

## [Decision Letter · Decision Letter 1]

24 Feb 2025

Analysis of structural and metabolic changes in surface microorganisms following powdery mildew infection in wheat and assessment of their potential function in biological control

PONE-D-24-24904R1

Dear Dr. Xue,

We’re pleased to inform you that your manuscript has been judged scientifically suitable for publication and will be formally accepted for publication once it meets all outstanding technical requirements.

Kind regards,

Kandasamy Ulaganathan

Academic Editor

PLOS ONE

Additional Editor Comments (optional):

Reviewers' comments:

Reviewer's Responses to Questions

**Comments to the Author**

1. If the authors have adequately addressed your comments raised in a previous round of review and you feel that this manuscript is now acceptable for publication, you may indicate that here to bypass the “Comments to the Author” section, enter your conflict of interest statement in the “Confidential to Editor” section, and submit your "Accept" recommendation.

Reviewer #2: All comments have been addressed

2. Is the manuscript technically sound, and do the data support the conclusions?

Reviewer #2: Yes

3. Has the statistical analysis been performed appropriately and rigorously? 

Reviewer #2: Yes

4. Have the authors made all data underlying the findings in their manuscript fully available?

Reviewer #2: Yes

5. Is the manuscript presented in an intelligible fashion and written in standard English?

Reviewer #2: Yes

6. Review Comments to the Author

Reviewer #2: Authors have modified the discussion and have adequately addressed my comments raised in a previous round of review. I have no more comments for the authors.

7. PLOS authors have the option to publish the peer review history of their article (what does this mean? ). If published, this will include your full peer review and any attached files.

**Do you want your identity to be public for this peer review?** For information about this choice, including consent withdrawal, please see our Privacy Policy .

Reviewer #2: **Yes: ** Latha Battu

---

## [Editor Report · Acceptance letter]

PONE-D-24-24904R1

PLOS ONE

Dear Dr. Xue,

I'm pleased to inform you that your manuscript has been deemed suitable for publication in PLOS ONE. Congratulations! Your manuscript is now being handed over to our production team.

Kind regards,

on behalf of

Dr. Kandasamy Ulaganathan

Academic Editor

PLOS ONE